

# Leaf morpho-physiological traits of *Populus sibirica* and *Ulmus pumila* in different irrigation regimes and fertilizer types

Ser-Oddamba Byambadorj[1,2], Jonathan Ogayon Hernandez[3], Sarangua Lkhagvasuren[1], Ge Erma[1], Khulan Sharavdorj[4], Byung Bae Park[2] and Batkhuu Nyam-Osor[1]

[1] Laboratory of Forest Genetics and Ecophysiology, National University of Mongolia, Ulaanbaatar, Mongolia
[2] Department of Environment and Forest Resources, College of Agriculture and Life Science, Chungnam National University, Deajeon, South Korea
[3] Department of Forest Biological Sciences, College of Forestry and Natural Resources, University of the Philippines, Los Baños, Philippines
[4] Crop Ecology Laboratory, College of Agriculture and Life Science, Chungnam National University, Deajeon, South Korea

Corresponding author
Batkhuu Nyam-Osor,
nbatkhuu@gmail.com

## ABSTRACT

**Background**. The impacts of climate change, such as increased soil dryness and nutrient deficiency, highlight the need for environmentally sustainable restoration of forests and groundwater resources. However, it is important to consider that extensive afforestation efforts may lead to a depletion of groundwater supply due to higher evapotranspiration rates, exacerbating water scarcity issues. Consequently, we conducted a study to examine how the fast-growing tree species *Populus sibirica* (Horth ex Tausch) and *Ulmus pumila* (L.) respond morpho-physiologically to varying watering regimes and types of fertilizers, aiming to better understand their specific water and nutrient requirements.

**Methods**. We used two-year-old nursery-growth seedlings ($N = 512$) of *P. sibirica* and *U. pumila* with initial root collar diameter (RCD) and the height of $0.51 \pm 0.02$ mm and $68 \pm 2.94$ cm and $0.33 \pm 0.01$ mm and $51 \pm 1.14$ cm, respectively. The leaf area (LA), specific leaf area (SLA), chlorophyll concentration, stomatal conductance ($g_s$), chlorophyll fluorescence, and predawn and midday leaf water potential were measured across treatments. Four different irrigation regimes and two different fertilizer types were applied: no irrigation (control, 0 L h$^{-1}$), 2 L h$^{-1}$ = 0.25 mm m$^{-2}$, 4 L h$^{-1}$ = 0.5 mm m$^{-2}$, 8 L h$^{-1}$ = 1.0 mm m$^{-2}$ and 120 g and 500 g tree$^{-1}$ of NPK and compost (COMP). Twelve plots (600 m$^2$) were established in the study site for each species and treatments.

**Results**. During the first growing season (2021), the LA of *P. sibirica* was larger in the 4–8 L h$^{-1}$ without fertilizer, but it was smaller in the 4 L h$^{-1}$+ COMP during the second growing season (2022). The 2 L h$^{-1}$ without fertilizer and 2 L h$^{-1}$ + NPK had larger LA compared with the control (CONT) for the first and second growing seasons, respectively, for *U. pumila*. *P. sibirica* seedlings at 4 L h$^{-1}$ without fertilizer had the highest SLA for 2021 and at 2 L h$^{-1}$ + NPK for 2022, whereas CONT and 4 L h$^{-1}$ had the highest SLA than the other treatments for 2021 and 2022 growing seasons, respectively, for *U. pumila*. The chlorophyll concentration of *P. sibirica* seedlings in the first year was

generally higher in CONT, while the 2 L h$^{-1}$ without any fertilizer yielded a significantly higher chlorophyll concentration of *U. pumila.* Chlorophyll fluorescence parameters (PI$_{ABS}$ and F$_m$) were generally lower in CONT with/without NPK or COMP for both species. The CONT with NPK/COMP generally had a higher g$_s$ compared with the other treatments in both experimental periods for *U. pumila,* whereas CONT and 2 L h$^{-1}$+ NPK-treated *P. sibirica* seedlings had a significantly greater g$_s$ during the first year and second year, respectively. The predawn and midday leaf water potentials of both species were generally the lowest in CONT, followed by 2 L h$^{-1}$+ NPK/COMP during the first growing season, but a different pattern was observed during the second growing season. Overall, the morpho-physiological traits of the two species were affected by watering and fertilizer treatments, and the magnitude of the effects varied depending on growing season, amount of irrigation, and fertilizer type, and their interactions.

## INTRODUCTION

According to climate model projections, the severity of "hot droughts" is expected to increase (*Cook, Ault & Smerdon, 2015*). Additionally, precipitation forecasts for semiarid regions are anticipated to remain uncertain (*Milly & Dunne, 2016*) in the coming decades. These environmental challenges have led to decreased precipitation and higher evapotranspiration rates, which significantly impact plant productivity in various parts of the world, especially arid and semi-arid regions (*Liang et al., 2017*; *Bista et al., 2018*). Apart from reducing plant growth and reproduction, drought has additional effects on certain plants, including decreased nutrient concentrations, changes in soil microbial activity (*Sanaullah et al., 2012*; *Bista et al., 2018*), reduced nutrient uptake per unit root, and decreased expression of nutrient-uptake proteins in roots (*Rouphael et al., 2012*). Consequently, understanding how terrestrial vegetation responds to drought remains a crucial challenge in arid and semi-arid regions, such as Mongolia. This complexity further complicates the implementation of environmental protection and restoration efforts in these areas.

Mongolia, being a semi-arid country, has encountered numerous urgent environmental challenges, including highly variable rainfall patterns on both intra- and interannual scales. These variations can be attributed to a combination of human activities and climate change (*Munkhzul et al., 2021*; *Unkelbach, Dulamsuren & Behling, 2021*). With the anticipated rise in temperatures due to climate change, soil dryness in Mongolia is expected to intensify (*Nandintsetseg et al., 2021*). Environmental restoration has been identified as a valuable and integrated solution to prolonged drought periods in Mongolia's arid, semi-arid, and dry sub-humid regions (*Cho et al., 2019*; *Byambadorj et al., 2020*; *Ser-Oddamba et al., 2020*). However, it is crucial to note that extensive afforestation efforts may lead to water depletion as a result of increased evapotranspiration rates. Consequently, arid and semi-arid countries may experience severe water scarcity (*Jackson et al., 2005*; *Krishnaswamy et*
*al., 2013*; *Hernandez, 2022*). Additionally, in tree plantations located in nutrient-limited arid and semi-arid areas, large-scale fertilizer application is commonly employed to enhance tree growth and productivity. However, different plant species have distinct nutrient requirements and fertility targets that depend on the prevailing environment and life-history traits of the species (*Salifu, Jacobs & Birge, 2009*). If the use of organic and inorganic fertilizers in tree plantations is not properly controlled, it can lead to pollution or contamination of groundwater (*Singh & Craswell, 2021*). Consequently, it is crucial to understand how tree species commonly utilized for environmental restoration respond to various watering regimes and fertilization practices in order to establish sustainable forest and groundwater resources in Mongolia.

Drought stress has adverse effects on plant growth and development, affecting both morphology and physiology (*Park et al., 2021*; *Wahab et al., 2022*). These effects are primarily linked to carbon allocation, which has implications for leaf carbon balance (*Körner, 2003*; *Allen, Breshears & McDowell, 2015*). Plants employ common morpho-physiological strategies in response to water deficits, including adjustments in stomatal conductance, leaf rolling, root-to-shoot ratio dynamics, transpiration efficiency, and delayed senescence (*Seleiman et al., 2021*). Various studies have reported significant reductions in leaf area, specific leaf area (SLA), pre-dawn and mid-day water potentials, as well as chlorophyll fluorescence parameters in several plant species as a response to drought (*Wellstein et al., 2017*; *Time, Garrido & Edmundo Acevedo, 2018*; *Smith et al., 2019*; *Shin et al., 2021*). However, most of the existing literature has primarily focused on the individual effects of drought stress on plant morpho-physiological traits, and there is limited information available regarding the combined effects of drought and other crucial factors (such as fertilizer application) on plant survival in semi-arid areas, particularly in tree species. The negative effects of drought stress on nutrient uptake have long been recognized, resulting in reduced nutrient inflow per unit of root length and biomass due to suppressed root growth (*Kuchenbuch, Claassen & Jungk, 1986*). Additionally, drought stress can significantly influence plant nutrient relations, leading to decreased nitrogen and phosphorus concentrations in plant tissues and altered nutrient uptake from the soil (*Ge et al., 2012*; *Bista et al., 2018*). However, while there are existing studies on this topic, the majority of findings primarily focus on herbaceous plants and grasses, leaving limited information available for other plant types.

Therefore, the present study aimed to examine the impact of different watering regimes and fertilizer types on the morpho-physiological characteristics of fast-growing tree species, specifically *Populus sibirica* (Horth ex Tausch) and *Ulmus pumila* (L.). These tree species have been widely employed in environmental restoration initiatives due to their adaptability to diverse environmental conditions, rapid re-sprouting ability, and high productivity (*Engelbrecht, Kursar & Tyree, 2005*; *Hu et al., 2008*; *Nyam-Osor et al., 2021*; *Montagnoli et al., 2022*). Despite their resilience to challenging environmental conditions, information regarding the optimal irrigation levels and fertilizer types for these species remains limited. Our expectation was that the leaf morpho-physiological traits of control seedlings (without irrigation or fertilizer) would be significantly affected compared to the treated seedlings subjected to various irrigation levels and fertilizers.

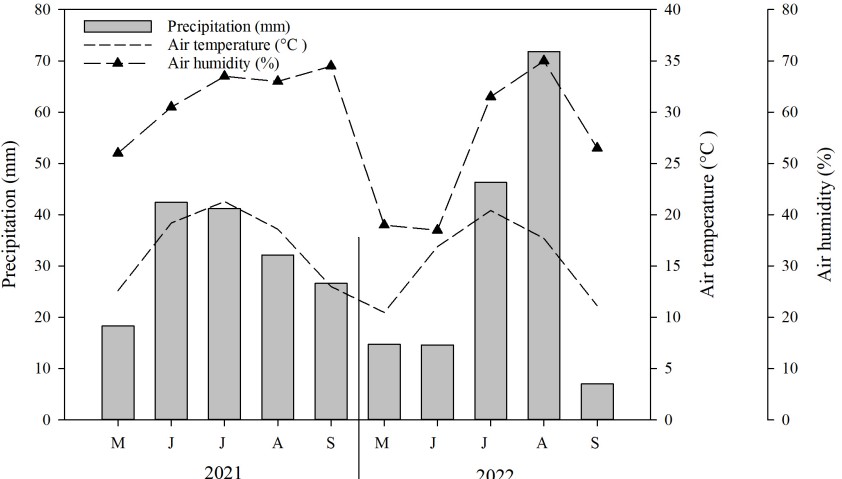

**Figure 1** Mean monthly precipitation (bar), air temperature (dotted line) and air humidity (broken, dotted line) in the experimental site during the conduct of the experiment (*i.e.*, May–September 2021–2022).

## MATERIALS AND METHODS

### Site description

The present study is one of the research outputs of a big project in Mongolia. Hence, the data were collected as previously described in our previous study (*Byambadorj et al., 2021b*). The study took place in Lun soum, Tuv Province, Mongolia, specifically located at latitude 47°52′15.43″N and longitude 105°10′46.4″E. Lun soum is situated approximately 135 kilometers west of Ulaanbaatar, and its elevation is 1,130 m above sea level. The research site is defined by the presence of Middle Khalkha dry steppe, an area that has undergone significant degradation due to extensive grazing activities by livestock (*Ulziykhutag, 1989*). The dominant vegetation in the area consists of xerophytic and mesoxerophytic grass species (*e.g.*, *Stipa sareptana* subsp. *krylovii* Roshev. and Cleistogenes squarrosa Trin. (*Lavrenko, Karamysheva & Nikulina, 1991*)). The study area is characterized by Kastanozems soil, specifically of the Loamic type, which is notable for its absence of distinct horizon differentiation (*IUSS Working Group WRB, 2015*). The surface soil and underlying soil hardness measure 4.5 kg cm$^{-2}$ and 1.7 kg cm$^{-2}$, respectively (*Batkhishig, 2016*).

Typically, the growing season in Mongolia commences in either May or June and concludes with the arrival of the initial frost in September (*NAMEM, 2021*). Throughout the duration of the study, the average temperature recorded was 0.5 °C, while the total rainfall ranged between 93.1 mm and 298.7 mm (2010–2021). Environmental data was gathered at the study site using the HOBO climate data logger. The data revealed that the highest average temperature (21.3 °C) and precipitation (71.8 mm) occurred during the months of July and August. Conversely, the lowest average temperature (10.5 °C) and precipitation (7 mm) were observed in May and September, respectively (Fig. 1).

## Experimental materials and design

This study was conducted in a 12-year-old of *Populus sibirica* and *Ulmus pumila* during two growing seasons (*i.e.,* July 2021 and 2022). We started the experiment in mid-May 2011 and started to measure leaf morpho-physiological traits from June to September of 2021 and 2022. Because the environment in Mongolia is quite varied, we were able to better identify seasonal trends or changes in the characteristics we analyzed by involving two growth seasons. In 2011, a total of 512 two-year-old seedlings were utilized for tree planting. These seedlings had initial root collar diameter (RCD) and height measurements of $0.51 \pm 0.02$ mm and $68 \pm 2.94$ cm, and $0.33 \pm 0.01$ mm and $51 \pm 1.14$ cm, respectively. The selected species possessed initial root lengths ranging from 21.50 to 25.9 cm. The planting process involved placing the seedlings into circular holes, which were approximately 60–70 cm deep and 50–60 cm wide. Furthermore, the seedlings received equal amounts of irrigation across all treatments during the acclimatization period, allowing them time to adapt to their new environment. Following the initial month, the study implemented four distinct irrigation regimes: a control group with no irrigation ($0 \text{ L h}^{-1}$), a group with irrigation at a rate of $2 \text{ L h}^{-1}$ ($0.25 \text{ mm m}^{-2}$), a group with irrigation at a rate of $4 \text{ L h}^{-1}$ ($0.5 \text{ mm m}^{-2}$), and a group with irrigation at a rate of $8 \text{ L h}^{-1}$ ($1.0 \text{ mm m}^{-2}$). Irrigation was applied every four days for a duration of five hours each time. An irrigation hose system connected to a 25-ton water tank facilitated the watering process.

Additionally, two types of fertilizers were used, namely NPK and compost (COMP), in different quantities. Prior to transplanting, these fertilizers were mixed with the natural soil to fill the holes. The NPK fertilizer was composed of solid granules containing a nitrogen, phosphorus, and potassium mixture with a ratio of 12:16:4. The quantities of fertilizers used were 120 g tree-1 for NPK and 500 g tree-1 for compost.

The compost used in the study was derived from thoroughly decomposed sheep manure, and it possessed the following chemical characteristics: pH of 7.4, organic matter content ranging from 18.0% to 25.0%, nitrogen content between $5.0 \text{ g kg}^{-1}$ and $7.0 \text{ g kg}^{-1}$, and total calcium (Ca), magnesium (Mg), potassium (K), and sodium (Na) contents of $9.29 \text{ g kg}^{-1}$, $7.02 \text{ g kg}^{-1}$, $9.18 \text{ g kg}^{-1}$, and $0.05 \text{ g kg}^{-1}$, respectively.To carry out the experiment, a total of twelve plots, each measuring 20 m × 10 m, were established for each tree species at the study site. These plots were divided as follows: four plots for the irrigation-only treatments, four plots for the irrigation combined with NPK fertilizer treatments, and four plots for the irrigation combined with compost treatments (Fig. 2). The plots were arranged with a spacing of 2.5 m between trees and 2.5 m between rows. For the irrigation-only treatments, a total of 32 seedlings were used, while for the irrigation combined with fertilizer or compost treatments, 16 trees were utilized.

## Measurements of leaf morphological traits

A total of 1,200 leaves, representing three replicates for each of the ten treatments and two tree species (20 leaves × three replicates × 10 treatments × two species), were collected for the determination of leaf area (LA) and specific leaf area (SLA). Leaves were not collected from the controlled plots as, by the end of the experiment, all seedlings in those plots were either dead or in a deteriorated condition. The collected leaf samples were initially

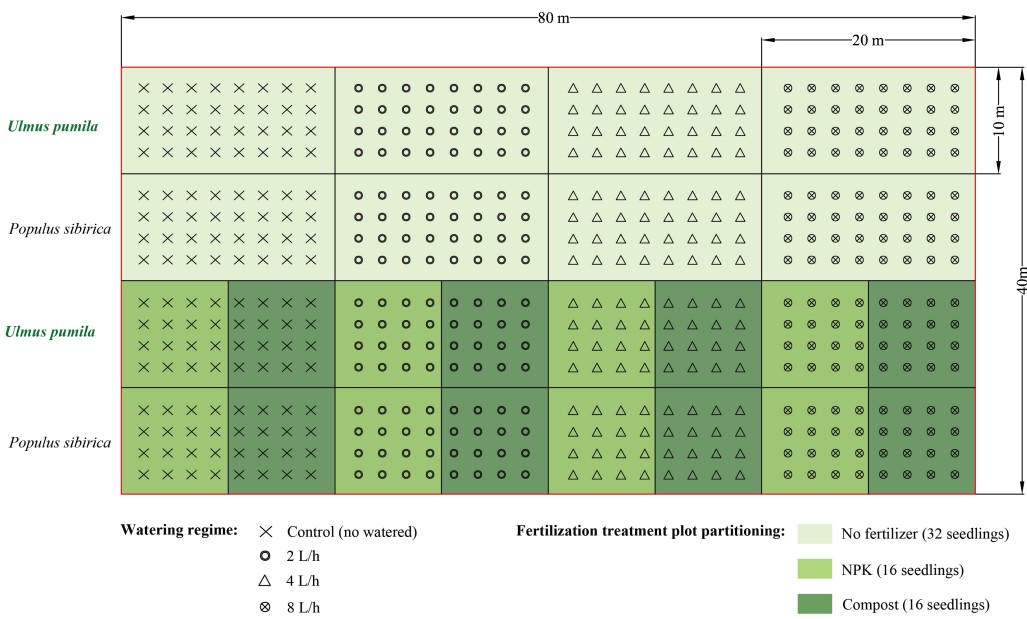

**Figure 2 Experimental design of the present study.**

placed in zip lock bags and stored in a cold container for transportation to the laboratory. Leaves were scanned using an HP LaserJet scanner (M1132 MFP, Palo Alto, CA, USA) at a resolution of 600 dpi. The image analysis software, ImageJ, was employed to analyze the leaf area (LA) based on the procedures outlined in *Schneider, Rasband & Eliceiri (2012)*, with modifications as described in *Hernandez et al. (2020)*. For the determination of specific leaf area (SLA), the leaves were dried in an oven at 65 °C for three days. Subsequently, the dried leaves were weighed using a high precision electronic scale ($d = 0.001$ g, Discovery Semi-Micro and Analytical Balance, Ohaus Corp., Switzerland). The SLA was calculated using the equation SLA = LA ($cm^2$) divided by the leaf dry mass (mg), following the procedure outlined by *Li, Shoup & Chen (2005)*.

## Measurements of leaf physiological traits
### Chlorophyll concentration
A random selection of 66 trees ($n = 9$) was made, and leaves that were healthy, fully expanded, and exposed to sunlight were harvested for the purpose of chlorophyll content analysis. The MC-100 leaf-clip chlorophyll meter from Apogee Instruments, Inc. (Logan, UT, USA) was used to perform the measurements. The chlorophyll content index (CCI), as described by *Padilla et al. (2018)*, was utilized for evaluating the chlorophyll levels.

### Stomatal conductance
The measurement of stomatal conductance involved using the SC-1 leaf porometer on sun-exposed leaves from the mid crown of 66 trees. The measurements were conducted between 08:00 and 11:00 h, with a sample size of nine trees. The SC-1 leaf porometer from Decagon Devices, Inc. (Pullman, WA, USA) was chosen for its capability to assess leaf conductance rates while considering the influence of surrounding environmental factors.

The calibration of the SC-1 leaf porometer ensured accurate and reliable measurements. To ensure accurate measurements, the calibration of the SC-1 leaf porometer was conducted before conducting field observations. This calibration was necessary as the leaf clip needed to reach temperature equilibrium with the surrounding environment. The readings from the leaf porometer were taken on the same leaf as the infrared thermometry (IRT) temperature readings to maintain consistency in the measurements. The SC-1 leaf porometer was used to measure leaf stomatal conductance automatically, providing readings in mmol m$^{-2}$ s$^{-1}$. The measurements were taken over a period of 30 s, during which the porometer also recorded air temperature and humidity. Stomatal conductance values near 0 mmol m$^{-2}$ s$^{-1}$ indicate significant stress, while values near 500 mmol m$^{-2}$ s$^{-1}$ suggest no stress on the leaves.

### Chlorophyll fluorescence

Chlorophyll a fluorescence measurement ($n = 9$) were performed on leaves collected from a total of 66 trees. The measurements were carried out on a sunny morning, between 08:00 and 11:00 h, prior to the midday depression of photosynthesis. The chlorophyll a fluorescence was quantified using a Handy PEA+ fluorimeter (Plant Efficiency Analyzer, Hansatech Instruments Ltd., Pentney, UK). This instrument allows for the measurement of changes in chlorophyll a fluorescence over a duration ranging from ten microseconds to one second. To prepare the leaves for measurement, a dark adaptation process was employed using specialized leaf clips. Following a dark-adapted period of 30 min, the leaves were exposed to saturating red light pulses with a peak wavelength of 650 nm and intensity of 3,000 $\mu$mol (photon) m$^{-2}$ s$^{-1}$. The signal gain for the season was set at $\times$ 1.0, as described in the works of *Strasser, Srivastava & Tsimilli-Michael (2000)* and *Yusuf et al. (2010)*. Table 1 provides a detailed description of the measurements related to the chlorophyll fluorescence OJIP transients.

## Leaf water potential

Using the methods outlined in *Scholander et al. (1964)*, the predawn ($\psi_p$) and midday ($\psi_m$) leaf water potentials were measured in healthy, fully expanded, and sun-exposed apical leaves. A pressure chamber (Model 1505D EXP, PMS Instrument Company, Albany, OR, USA) was employed for this purpose. The measurements were taken at a height of approximately 1.5 m above the ground.

## Statistical analysis

To evaluate the impact of irrigation and fertilization treatments on various morpho-physiological traits, such as chlorophyll concentration, stomatal conductance, chlorophyll fluorescence, and leaf water potential, a one-way to three-way analysis of variance (ANOVA) was conducted for each species. To further analyze and compare the treatments, Duncan's multiple range test (DMRT) was employed. All statistical analyses were performed using the Statistical Analysis Software (SAS) package developed by SAS Institute Inc. in 2014.

**Table 1** Summary of the OJIP-test parameters using data extracted from the fast fluorescence transient.

| | Fluorescence parameter | Calculation |
|---|---|---|
| Fo | Minimal fluorescence when all photosystem II reaction centers are open (at 50millisecond) | – |
| Fj | Fluorescence intensity at the J-step (at 2 millisecond) | – |
| Fi | Fluorescence intensity at the I-step (at 30 millisecond) | – |
| Fm (Fp) | Maximal fluoresxence intensity when all photosystem II reaction centers are closed | – |
| Vj | Relative variable Chl fluorescence (at 2 millisecond) | $(FJ - Fo)/(FM - Fo)$ |
| Vi | Relative variable Chl fluorescence (at 30 millisecond) | $(FI - Fo)/(FM - Fo)$ |
| Fv/Fm | Maximum quantum yield of PSII photochemistry measured in the dark-adapted state | $(Fm - F0)/Fm$ |
| $PI_{ABS}$ | Performance index on an absorption basis: the efficiency of energy conservation from absorbed photons to a reduction in intersystem electron carriers | $(RC/ABS)\ [(\Phi Po/(1 - \Phi Po)]\ [(1 - VJ)/(1 - (1 - VJ))]$ |
| Mo | Slope at the beginning of the transient $FO \rightarrow FM$, maximal fractional rate of photochemistry approximated initial slope (per millisecond) of the fluorescence transient V = f(t). This parameter reflects the rate at which photosystem II reaction centers are closed. It is a net value, because reduced QA can be reoxidized as a result of electron transport outside QA. | $4\ (F300 - Fo)/(FM - Fo)$ |
| Ss | Normalized total area above the OJIP curve (reflecting multiple turnover of QA reductions). The Ss minimum appears only when each QA is reduced only once the smallest SM turn-over (single turn-over) | $Area/(FM - Fo)$ |
| | Specific fluxes per active PSII reaction center | |
| ABS/RC | Average absorbed photon flux per photosystem II reaction center | $Mo\ (1/VJ)\ (1/\ \Phi Po)$ |
| TRo/RC | Energy flux trapped by one active photosystem II reaction center | $Mo\ (1/VJ)$ |
| ETo/RC | Rate of electron transport flux from QA to QB per photosystem II reaction center | $Mo\ (1/VJ)\ (1 - VJ)$ |
| DIo/RC | Energy flux not intercepted by a reaction center | $ABS/RC - TRo/RC$ |
| REo/RC | Electron flux leading to a reduction in the PSI end acceptor | $Mo\ (1/VJ)\ (1 - VI)$ |
| $\Phi Po$ | Quantum yield of primary photochemistry; probability that an absorbed photon leads to a reduction in QA | $(FM - Fo)/FM$ |

# RESULTS

## Effects different irrigation regimes and fertilizer types on leaf morphological traits of *Populus sibirica* and *Ulmus pumila*

Results of the present study revealed significant variation in leaf area (LA) between control and NPK/COMP-treated seedlings of *P. sibirica* and *U. pumila* (Table 2; Table S1). As expected, the control group generally yielded smaller LA compared with NPK/COMP-treated seedlings, suggesting that control seedlings' leaf growth were negatively affected. The effects of the treatments also differed by growing season. In *P. sibirica*, the 4–8 L h$^{-1}$ without fertilizer had larger LA than control (CONT), NPK, and COMP treatments, which all had similar effects on LA, during the first growing season. A different pattern in LA was

**Table 2 Leaf area (LA) and specific leaf area (SLA) of *Populus sibirica* and *Ulmus pumila* across different treatments measured in July, 2021 and July, 2022.** Different lowercase letters within each species indicate significant differences among the treatments at $\alpha = 0.05$. Vertical bars represent standard errors.

| Species | Treatments | LA | | SLA | |
|---|---|---|---|---|---|
| | | July, 2021 | July, 2022 | July, 2021 | July, 2022 |
| | CONT | $8.21 \pm 2.23^{ab}$ | $6.98 \pm 0.38^{d}$ | $26.20 \pm 0.62^{ab}$ | $24.16 \pm 0.72^{c}$ |
| | 2 L h$^{-1}$ | $9.52 \pm 2.14^{ab}$ | $9.30 \pm 0.76^{b}$ | $25.40 \pm 0.87^{ab}$ | $26.75 \pm 0.54^{ab}$ |
| | 4 L h$^{-1}$ | $10.69 \pm 2.86^{a}$ | $8.67 \pm 0.42^{bc}$ | $27.95 \pm 1.11^{a}$ | $25.79 \pm 0.60^{bc}$ |
| | 8 L h$^{-1}$ | $10.16 \pm 2.62^{a}$ | $7.64 \pm 0.41^{cd}$ | $26.43 \pm 0.87^{ab}$ | $26.82 \pm 0.68^{ab}$ |
| | NPK | – | – | – | – |
| *Populus* | 2 L h$^{-1}$+NPK | $6.07 \pm 2.02^{b}$ | $8.64 \pm 0.67^{bc}$ | $25.99 \pm 0.29^{ab}$ | $28.65 \pm 1.11^{a}$ |
| *sibirica* | 4 L h$^{-1}$+NPK | $8.81 \pm 1.95^{ab}$ | $7.08 \pm 0.25^{d}$ | $26.17 \pm 0.37^{ab}$ | $25.03 \pm 0.48^{bc}$ |
| | 8 L h$^{-1}$+NPK | $9.91 \pm 1.78^{ab}$ | $9.27 \pm 0.51^{b}$ | $26.21 \pm 0.58^{ab}$ | $25.57 \pm 0.31^{bc}$ |
| | COMP | – | – | – | – |
| | 2 L h$^{-1}$+COMP | $8.05 \pm 3.28^{ab}$ | $8.17 \pm 0.48^{bcd}$ | $24.71 \pm 0.34^{b}$ | $25.35 \pm 0.88^{bc}$ |
| | 4 L h$^{-1}$+COMP | $8.68 \pm 2.78^{ab}$ | $10.81 \pm 0.20^{a}$ | $23.91 \pm 0.40^{b}$ | $24.29 \pm 0.45^{c}$ |
| | 8 L h$^{-1}$+COMP | $8.91 \pm 3.13^{ab}$ | $8.83 \pm 0.41^{bc}$ | $26.12 \pm 1.71^{ab}$ | $25.03 \pm 0.42^{bc}$ |
| | CONT | $4.04 \pm 0.19^{c}$ | $2.55 \pm 0.18^{c}$ | $24.07 \pm 0.81^{a}$ | $26.21 \pm 0.61^{abc}$ |
| | 2 L h$^{-1}$ | $6.86 \pm 0.21^{a}$ | $3.17 \pm 0.41^{bc}$ | $19.81 \pm 0.94^{cd}$ | $22.36 \pm 0.46^{bcde}$ |
| | 4 L h$^{-1}$ | $5.62 \pm 0.45b$ | $2.81 \pm 0.30^{bc}$ | $18.21 \pm 0.36^{ef}$ | $28.33 \pm 5.80^{a}$ |
| | 8 L h$^{-1}$ | $3.02 \pm 0.26^{cd}$ | $2.38 \pm 0.17^{cd}$ | $21.89 \pm 0.15^{b}$ | $20.13 \pm 0.60^{def}$ |
| | NPK | $2.73 \pm 0.37^{d}$ | $1.70 \pm 0.15^{d}$ | $21.12 \pm 0.37^{bc}$ | $26.39 \pm 0.41^{ab}$ |
| *Ulmus* | 2 L h$^{-1}$+NPK | $5.23 \pm 0.21^{b}$ | $4.08 \pm 0.34^{a}$ | $21.06 \pm 0.27^{bc}$ | $24.17 \pm 0.46^{abcd}$ |
| *pumila* | 4 L h$^{-1}$+NPK | $6.12 \pm 0.63^{ab}$ | $3.49 \pm 0.35^{ab}$ | $19.15 \pm 0.27^{de}$ | $19.36 \pm 0.31^{def}$ |
| | 8 L h$^{-1}$+NPK | $5.57 \pm 0.40^{b}$ | $3.15 \pm 0.15^{bc}$ | $17.38 \pm 0.72^{f}$ | $17.01 \pm 0.36^{ef}$ |
| | COMP | $5.72 \pm 0.36^{b}$ | $2.49 \pm 0.16^{c}$ | $21.99 \pm 0.41^{b}$ | $24.59 \pm 1.41^{abcd}$ |
| | 2 L h$^{-1}$+COMP | $3.61 \pm 0.19^{cd}$ | $2.50 \pm 0.11^{c}$ | $21.47 \pm 0.31^{b}$ | $23.34 \pm 0.57^{abcd}$ |
| | 4 L h$^{-1}$+COMP | $2.77 \pm 0.27^{d}$ | $2.87 \pm 0.22^{bc}$ | $14.88 \pm 0.37^{g}$ | $16.45 \pm 0.23^{f}$ |
| | 8 L h$^{-1}$+COMP | $5.57 \pm 0.48^{b}$ | $3.58 \pm 0.15^{ab}$ | $20.54 \pm 0.24^{bcd}$ | $20.52 \pm 1.47^{cdef}$ |

observed during the second growing season, *i.e.,* 4 L h$^{-1}$ + COMP resulted in the highest LA compared with the other treatments. In the case of *U. pumila*, the 2 L h$^{-1}$ without fertilizer and 2 L h$^{-1}$ + NPK had larger LA compared with CON and other NPK/COMP treatments for 2021 and 2022 growing seasons, respectively. Moreover, the results of our study have revealed an intriguing interaction effect between year, irrigation, and fertilization on LA for both *P. sibirica* and *U. pumila* (Table 2; Table S2). Over time, such a combination resulted in a substantially worse effect in the LA in the control groups lacking irrigation and fertilizer.

The SLA also varied significantly between control and NPK/COMP-treated seedlings of *P. sibirica* and *U. pumila*, and the pattern differed by growing season (Table 2; Table S2). We also found a significant egffect of year × irrigation × fertilization interaction on SLA of *U. pumila* (Table S2). Among treatments, *P. sibirica* seedlings at 4 L h$^{-1}$ without fertilizer had the highest SLA for 2021 and at 2 L h$^{-1}$ + NPK for 2022. A different pattern was observed in *U. pumila, i.e.,* CONT and 4 L h$^{-1}$ had the highest SLA than the other treatments for 2021 and 2022 growing seasons, respectively.

## Effects different irrigation regimes and fertilizer types on leaf physiological traits of *Populus sibirica* and *Ulmus pumila*

The chlorophyll concentration, stomatal conductance ($g_s$), and chlorophyll fluorescence varied significantly between treatments in both *P. sibirica* and *U. pumila* (Figs. 3–6; Table 2; Table S3). In this study, we found no significant effects of year × irrigation × fertilization interaction on leaf physiological traits of both species, except in stomatal conductance of *U. pumila* (Tables S4; S5). However, the year × fertilization interaction significantly affected $F_m$, $F_v$, $F_v/F_m$, and $PI_{ABS}$ in both species (Table S5). The effects of irrigation × fertilization interaction on the chlorophyll fluorescence parameters, except in $F_m$, were also significant in the case of *U. pumila*. In *P. sibirica*, the chlorophyll concentration in the first year was generally higher in CONT, watering-alone treatments, and 2 L h$^{-1}$+NPK compared with the other treatments (Fig. 3). This contradicts our hypothesis that the leaf morpho-physiological characteristics of control seedlings would be significantly worse than those of treated seedlings. A different pattern was observed in *U. pumila*, *i.e.,* the 2 L h$^{-1}$ without any fertilizer yielded a significantly higher chlorophyll concentration compared with other treatments during the second year of the experiment. Notably, the NPK-alone treatment in *U. pumila* had the lowest chlorophyll concentration among the treatments.

The stomatal conductance ($g_s$) varied significantly between treatments, and the pattern was different by year in both *P. sibirica* and *U. pumila* (Fig. 4). In *P. sibirica*, CONT had a significantly greater $g_s$ than the other treatments during the first year. A contrasting pattern was observed during the second year of the experiment, *i.e.,* the highest $g_s$ was found in 2 L h$^{-1}$+NPK, while the lowest values were generally recorded in CONT, all watering-alone treatments, and in 4 L h$^{-1}$+ COMP. In the case of *U. pumila*, the CONT with NPK or COMP generally had a higher $g_s$ compared with the other treatments in both experimental periods. The lowest values were generally recorded in 8 L h$^{-1}$+ NPK in both experimental periods, although all water + fertilizer treatment combinations decreased relative to CONT with and without NPK or COMP.

The whole OJIP curves of *P. sibirica* and *U. pumila* under different watering and fertilization treatments are shown in Figs. 5 and 6. Here, we found different O-J-I-P kinetic curves patterns for only water treatments, water + NPK treatments, and water + COMP treatments for both species and experimental periods. The general pattern is that the fluorescence transients at CONT were characterized by lower values of curves than those at either NPK/COMP-treated ones. Similar O (origin) phase curve (minimal fluorescence) was observed across treatments and species in both experimental periods. In the P (peak) phase of *P. sibirica*, the 2 L h$^{-1}$ generally had the highest fluorescence, while the lowest was observed in CONT for only water treatments, water + NPK treatments, and water + COMP treatments, particularly in 2021 experimental period. Different patterns were observed in *U. pumila* depending on treatments. For only water treatments, 2 L h$^{-1}$ generally had the highest fluorescence in both experimental periods. The CONT and/or 2 L h$^{-1}$ + NPK yielded the highest fluorescence in 2021, but a different pattern was observed in 2022 (*i.e.,* 2 L h$^{-1}$ + NPK = 4 L h$^{-1}$ + NPK = 8 L h$^{-1}$ + NPK > CONT) for water + NPK treatments. A contrasting pattern was observed in the case of water + COMP treatments, *i.e.,* 4 L

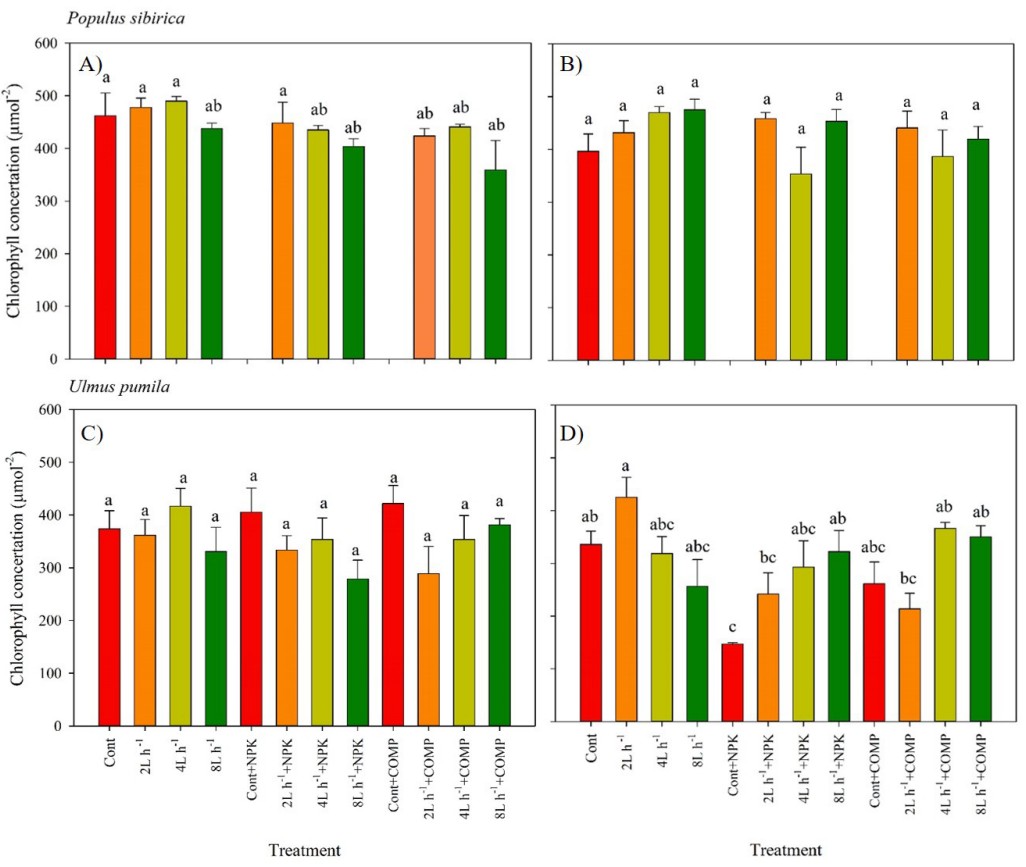

**Figure 3 Chlorophyll concentration of *Populus sibirica* and *Ulmus pumila* in different treatments measured in (A) July, 2021, (B) July, 2022, (C) July, 2021, (D) July, 2022.** Different lowercase letters within each species indicate significant differences between treatments at $\alpha = 0.05$. Vertical bars represent standard errors.

h$^{-1}$+ COMP and/or 8 L h$^{-1}$+ COMP generally had the highest fluorescence among the treatments for *U. pumila*.

We observed changes in chlorophyll a fluorescence transient parameters across different treatments, and the patterns differed by year depending on the type of fertilizer and amount of irrigation (Figs. S1; S2). In both species, no significant changes in other parameters (*e.g.*, $F_v/F_m$, $F_o$, TR$_o$/RC) were detected during the study periods (Table 3). Here, PI$_{ABS}$ and $F_m$ were generally lower in CONT with/without NPK or COMP for both *P. sibirica* and *U. pumila* (Table 3). In *P. sibirica*, PI$_{ABS}$ and $F_m$ were generally greater in 2 L h$^{-1}$ + NPK or COMP throughout the first experimental period. Different patterns in the treatment with the highest value were detected throughout the second year for both PI$_{ABS}$ (*i.e.*, only 8 L h$^{-1}$, 8 L h$^{-1}$ + NPK, 2 L h$^{-1}$ + COMP) and $F_m$ (*i.e.*, only 2 L h$^{-1}$, 2 L h$^{-1}$+ NPK, 2/4/8 L h$^{-1}$+ COMP).

The effects of applied irrigation and fertilizer on diurnal leaf water potential are shown in Figs. 7–8. Among the treatments, predawn and midday leaf water potentials of *P. sibirica* and *U. pumila* were generally the lowest in CONT, followed by 2 L h$^{-1}$+NPK/COMP

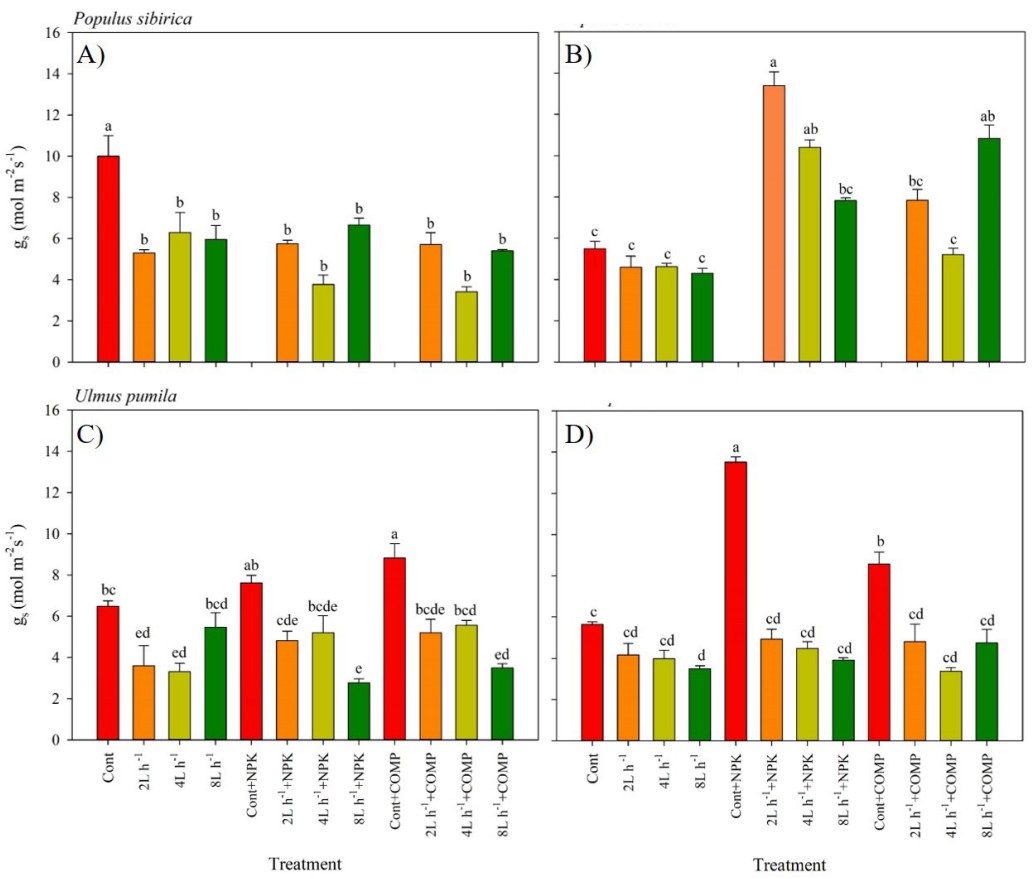

**Figure 4 Stomatal conductance of *Populus sibirica* and *Ulmus pumila* in different treatments measured in (A) July, 2021, (B) July, 2022, (C) July, 2021, (D) July, 2022.** Different lowercase letters within each species indicate significant differences between treatments at $\alpha = 0.05$. Vertical bars represent standard errors.

during the first growing season. For *P. sibirica* during the first year of the experiment, leaf water potential began to drop from 5:00 to 13:00 (midday), especially for water + COMP treatments, and began to increase as early as 9:00 to 23:00 for only water and water + NPK treatments. During the second growing season for *P. sibirica,* an obviously different pattern was observed, *i.e.,* the 8 L h$^{-1}$+COMP had the lowest midday leaf water potential among the treatments. For *U. pumila* during the second growing season, leaf water potential at only NPK treatment had declined much lower (more negative) than only water or only COMP treatments, although it showed a tendency to increase starting from 19:00.

The pattern of treatment effects on predawn ($\psi_p$) and midday ($\psi_m$) leaf water potentials of *P. sibirica* and *U. pumila* differed between 2021 and 2022 growing seasons (Table 4). In *P. sibirica*, CONT generally had a significantly lower (more negative) $\psi_p$ and $\psi_m$ compared with NPK + 2L h$^{-1}$/4L h$^{-1}$ irrigation, particularly in the second growing season. The water + NPK treatments showed a tendency to have a higher $\psi_p$ and $\psi_m$ than water + COMP treatments, regardless of the amount of irrigation for *P. sibirica*. The $\psi_p$ was generally the lowest (more negative) at NPK without water among NPK and COMP treatments for

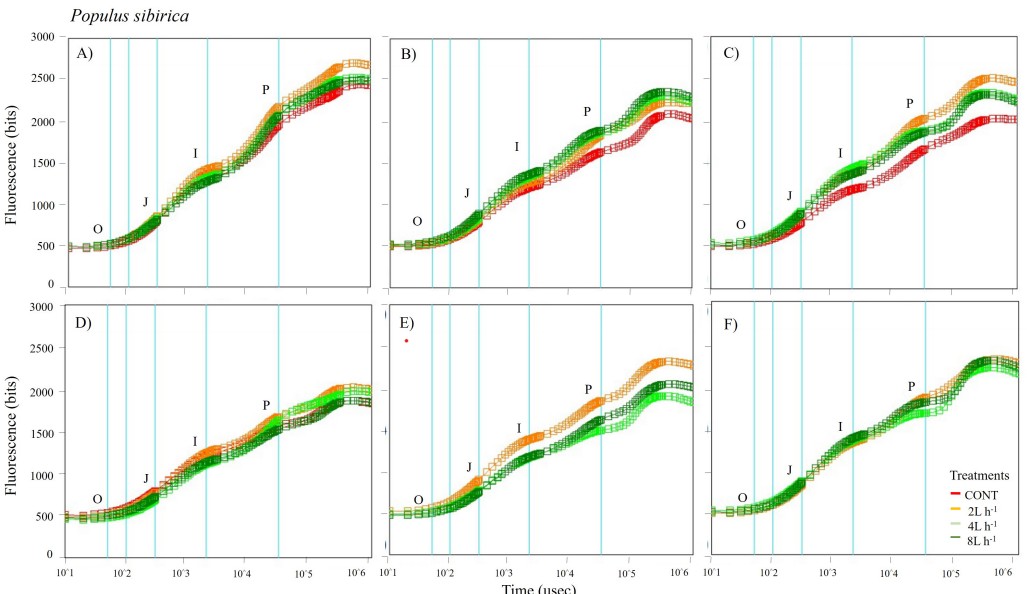

**Figure 5** Chlorophyll fluorescence OJIP kinetic curves of *Populus sibirica* in different treatments measured **(A)** only water treatments **(B)** water+NPK treatments **(C)** water+COMP treatments in July, 2021 and **(D)** only water treatments **(E)** water+NPK treatments **(F)** water+COMP in July, 2022. Different lowercase letters within each species indicate significant differences between treatments at $\alpha = 0.05$. Vertical bars represent standard errors.

*U. pumila.* Lastly, the $\psi_m$ of *U. pumila* was the highest in either NPK or COMP alone treatments than NPK/COMP + water treatments.

## DISCUSSION

### Effects of different fertilization and watering regimes on the leaf area and specific leaf area of *Populus sibirica* and *Ulmus pumila*

As expected, the leaves of *P. sibirica* displayed notable responses to varying fertilization and watering conditions, aligning with the findings of previous studies (*Larwanou, Adamou & Abasse, 2014*; *Guo et al., 2021*). In the initial growing season, significantly higher leaf area (LA) and specific leaf area (SLA) were observed in the 4 L h$^{-1}$ and/or 8 L h$^{-1}$ without fertilizer treatments compared to the control (CONT) and other treatment groups. This suggests that the presence of water alone may have been sufficient for stimulating leaf growth in *P. sibirica*, without the need for additional NPK or COMP fertilizers. Given the semi-arid nature of the study site, it is possible that *P. sibirica* seedlings initially prioritized energy investment in expanding leaf area by utilizing water resources more efficiently during the early stages of development. This preference for water over fertilizer may be attributed to the higher water and energy requirements associated with the absorption and uptake of dissolved nutrients *via* mass flow, resulting in lower leaf area as observed in seedlings treated with water in combination with NPK or COMP, particularly in the first growing season. Furthermore, during the initial growth and development stages, cells typically possess meristematic properties and have higher water demands compared to

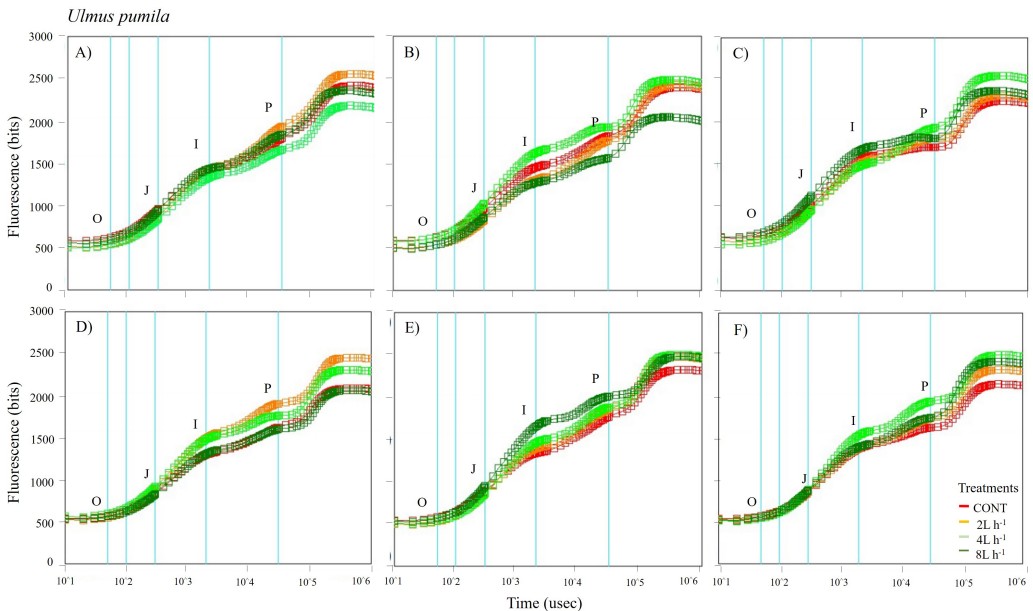

**Figure 6** Chlorophyll fluorescence OJIP kinetic curves of *Ulmus pumila* in different treatments measured **(A)** only water treatments **(B)** water+NPK treatments **(C)** water+COMP treatments in July, 2021 and **(D)** only water treatments **(E)** water+NPK treatments **(F)** water+COMP in July, 2022. Different lowercase letters within each species indicate significant differences between treatments at $\alpha = 0.05$. Vertical bars represent standard errors.

fertilizer requirements. The process of plant cell expansion involves the concurrent uptake of water into the cells (*Cosgrove, 2014*). This phenomenon can help explain the significant impact of the interaction between year, irrigation, and fertilization on the leaf area (LA) of *P. sibirica* seedlings. During the initial stages of growth, there was a greater demand for irrigation compared to fertilizer, which may have intensified as the seedlings continued to develop. The need for fertilizer could have arisen from a gradual increase in growth over time. This finding further elucidates the lower LA observed in the control group (no irrigation), as water likely played a crucial role in stimulating leaf area growth.

However, a different trend was observed in *U. pumila* seedlings, where a lower water amount (2 L h$^{-1}$) resulted in higher leaf area (LA) compared to the other treatments during the first growing season. This suggests that *U. pumila* has a lower water requirement for enhancing leaf area growth compared to *P. sibirica*, despite both species being known for their drought resistance. The potentially higher drought resistance of *U. pumila* can be inferred from the significant effect of the interaction between year, irrigation, and fertilization on LA and specific leaf area (SLA) of *P. sibirica*. Additionally, *U. pumila* is classified as an invasive noxious tree, capable of reproducing through wind-dispersed seeds and sprouts, and able to thrive in regions where other trees struggle, even in dry mesic areas (*USDA, 2017*). It is possible that *P. sibirica* seedlings had already adapted to the prevailing environment after the first growing season, leading to a contrasting pattern in LA (*i.e.,* 4 L h$^{-1}$ + COMP > CONT) and SLA (*i.e.,* 2 L h$^{-1}$ + NPK > CONT) during the second growing

**Table 3  Chlorophyll fluorescence parameters (photosystem II (Fv/Fm), performance index ($PI_{ABS}$) of *Populus sibirca* and *Ulmus pumila* in different treatments measured in July, 2021 and July, 2022.**  Different lowercase letters within each species indicate significant differences between treatments at $\alpha = 0.05$. Vertical bars represent standard errors.

| Species | Treatments | $F_0$ | | $F_m$ | | $F_v$ | | $F_v/F_m$ | | $PI_{ABS}$ | |
|---|---|---|---|---|---|---|---|---|---|---|---|
| | | July, 2021 | July, 2022 | July, 2021 | July, 2022 | July, 2021 | July, 2022 | July, 2021 | July, 2022 | July, 2021 | July, 2022 |
| *Populus sibirica* | CONT | 443.16 ± 5.16c | 494 ± 16.43c | 2451.16 ± 125.9abc | 1860.77 ± 57.85d | 2008 ± 127.39abcd | 1356.77 ± 51.33d | 0.816 ± 0.009ab | 0.728 ± 0.009f | 3.34 ± 0.46ab | 1.15 ± 0.14d |
| | 2 L h$^{-1}$ | 461.11 ± 6.1bc | 504.88 ± 22.75c | 2703.66 ± 68.78a | 2024.88 ± 65.48bcd | 2242.55 ± 68.49a | 1530 ± 61.49bcd | 0.828 ± 0.01a | 0.754 ± 0.011cde | 3.87 ± 0.21a | 1.66 ± 0.22cd |
| | 4 L h$^{-1}$ | 462.77 ± 9.76c | 440.77 ± 10.29c | 2535 ± 88.63ab | 1984.66 ± 51.01cd | 2072.22 ± 90.21ab | 1543.88 ± 57.63bcd | 0.815 ± 0.004ab | 0.776 ± 0.009abc | 3.644 ± 0.31a | 2.77 ± 0.22c |
| | 8 L h$^{-1}$ | 469.11 ± 9.24bc | 476.33 ± 19.5bc | 2512.88 ± 75.96abc | 1923.77 ± 89.55cd | 2043.77 ± 80.2abc | 1447.44 ± 80.82cd | 0.811 ± 0.009ab | 0.750 ± 0.009def | 3.93 ± 0.38a | 1.73 ± 0.15abc |
| | NPK | – | – | – | – | – | – | – | – | – | – |
| | 2 L h$^{-1}$+NPK | 491.66 ± 7.58c | 508.66 ± 21.56bc | 2007 ± 94.14c | 2359.88 ± 76.23a | 1515.33 ± 91.32d | 1851.22 ± 64.59a | 0.78 ± 0.010bc | 0.784 ± 0.007ab | 2.38 ± 0.27bc | 2.011 ± 0.22bc |
| | 4 L h$^{-1}$+NPK | 482.33 ± 11.11bc | 540.66 ± 34.21a | 2214 ± 99.05bc | 2094.66 ± 104.88bc | 1731.66 ± 93.07bcd | 1554 ± 75.27bc | 0.75 ± 0.006c | 0.742 ± 0.006ef | 1.96 ± 0.24c | 1.276 ± 0.18d |
| | 8 L h$^{-1}$+NPK | 487 ± 10.44bc | 459.25 ± 6.91bc | 2345.88 ± 135.09b | 2058.37 ± 53.91bcd | 1858.88 ± 127.14bcd | 1599.12 ± 52.62bcd | 0.78 ± 0.007bc | 0.776 ± 0.006abc | 2.31 ± 0.13bc | 22.324 ± 0.23ab |
| | COMP | – | – | – | – | – | – | – | – | – | – |
| | 2 L h$^{-1}$+COMP | 538.55 ± 18.51abc | 470.11 ± 9.08bc | 2295.88 ± 68.84abc | 2338 ± 63.76a | 1757.33 ± 85.21abcd | 1867.88 ± 64.057a | 0.81 ± 0.006ab | 0.79 ± 0.007a | 2.74 ± 0.33abc | 2.47 ± 0.27ab |
| | 4 L h$^{-1}$+COMP | 496 ± 9.91bc | 509.77 ± 10.19ab | 2372.22 ± 137.39bc | 2207.11 ± 58.18ab | 1876.22 ± 139.94cd | 1697.33 ± 57.35ab | 0.76 ± 0.007c | 0.77 ± 0.008bcd | 1.67 ± 0.31c | 1.75 ± 0.35cd |
| | 8 L h$^{-1}$+COMP | 494 ± 11.53bc | 476.88 ± 7.04bc | 2315 ± 66.71bc | 2305.88 ± 55.81a | 1821 ± 71.62bcd | 1829 ± 56.46a | 0.78 ± 0.009bc | 0.79 ± 0.005ab | 2.12 ± 0.28c | 2.19 ± 0.27abc |
| *Ulmus pumila* | CONT | 562.14 ± 13.74ab | 506.44 ± 20.37ab | 2424.28 ± 80.31ab | 2048.88 ± 153.92c | 1862.14 ± 77.41abc | 1469.11 ± 162.31d | 0.76 ± 0.008bc | 0.71 ± 0.034ab | 1.75 ± 0.27bcde | 1.56 ± 0.42a |
| | 2 L h$^{-1}$ | 483.88 ± 6.92a | 530.56 ± 14.15cd | 2562.33 ± 87.57a | 2362.33 ± 97.42abc | 2078.44 ± 85.53a | 1855.88 ± 108.68abc | 0.81 ± 0.004a | 0.78 ± 0.014bc | 3.01 ± 0.19a | 2.05 ± 0.46b |
| | 4 L h$^{-1}$ | 479.25 ± 57.78ab | 569.55 ± 15.02ab | 2187.75 ± 279.32ab | 2209.11 ± 73.92abc | 1708.5 ± 22.012ab | 1639.55 ± 84.369bcd | 0.78 ± 0.014abc | 0.74 ± 0.014abc | 1.94 ± 0.17bcd | 1.21 ± 1.46ab |
| | 8 L h$^{-1}$ | 548.88 ± 23.67ab | 513.66 ± 9.59cd | 2355.44 ± 77.45ab | 2073 ± 61.82c | 1806.55 ± 86.88abc | 1559.33 ± 66.44cd | 0.77 ± 0.009bcd | 0.75 ± 0.009ab | 1.78 ± 0.21bcde | 1.59 ± 0.09ab |
| | NPK | 571.88 ± 11.22ab | 517.89 ± 9.89bcd | 2372.11 ± 72.15ab | 2330.33 ± 125.66abc | 1800.22 ± 78.76abc | 1812.44 ± 124.92abc | 0.75 ± 0.010bcd | 0.77 ± 0.012ab | 1.63 ± 0.45bcdef | 2.19 ± 0.31ab |
| | 2 L h$^{-1}$+NPK | 645.55 ± 34.34ab | 493.55 ± 16.44d | 2253.22 ± 30.23ab | 2415 ± 128.61ab | 1607.66 ± 39.05c | 1921.44 ± 136.51ab | 0.77 ± 0.010bc | 0.78 ± 0.016ab | 2.33 ± 0.16ab | 2.39 ± 0.45a |
| | 4 L h$^{-1}$+NPK | 539.33 ± 31.12b | 494.77 ± 14.39d | 2377.66 ± 54.41b | 2437.556 ± 61.67ab | 1838.33 ± 79.19abc | 1942.77 ± 68.72ab | 0.74 ± 0.007cd | 0.79 ± 0.010a | 1.11 ± 0.21def | 2.35 ± 0.39a |
| | 8 L h$^{-1}$+NPK | 574.33 ± 13.04b | 497.55 ± 16.18d | 2264.33 ± 83.74b | 2489.88 ± 57.63a | 1690 ± 92.16bc | 1992.33 ± 53.11a | 0.75 ± 0.009bc | 0.79 ± 0.006a | 1.33 ± 0.32cdef | 1.917 ± 0.26ab |
| | COMP | 570.11 ± 19.06b | 535 ± 17.33abcd | 2268.55 ± 113.05b | 2150.44 ± 45.05bc | 1698.44 ± 111.03c | 1615.44 ± 50.73bcd | 0.72 ± 0.014d | 0.75 ± 0.010abc | 0.93 ± 0.14ef | 1.46 ± 0.46a |
| | 2 L h$^{-1}$+COMP | 567 ± 35.81b | 504.33 ± 7.86cd | 2460 ± 97.32b | 2274.44 ± 95.16abc | 1893 ± 125.88bc | 1770.11 ± 99.08abcd | 0.743 ± 0.014cd | 0.77 ± 0.010a | 1.19 ± 0.07cdef | 2.04 ± 0.46a |
| | 4 L h$^{-1}$+COMP | 492.22 ± 53.87ab | 492.88 ± 10.17d | 2054.44 ± 256.31ab | 2481 ± 75.83a | 1562.22 ± 207.84ab | 1988.11 ± 81.21a | 0.783 ± 0.018ab | 0.79 ± 0.009a | 2.05 ± 0.18bc | 2.28 ± 0.42ab |
| | 8 L h$^{-1}$+COMP | 615.44 ± 13.01ab | 577.77 ± 27.57a | 2407.11 ± 44.08ab | 2051.22 ± 143.37c | 1791.66 ± 40.41abc | 1465.44 ± 157.42d | 0.744 ± 0.005cd | 0.69 ± 0.030c | 0.81 ± 0.16f | 1.34 ± 1.31b |

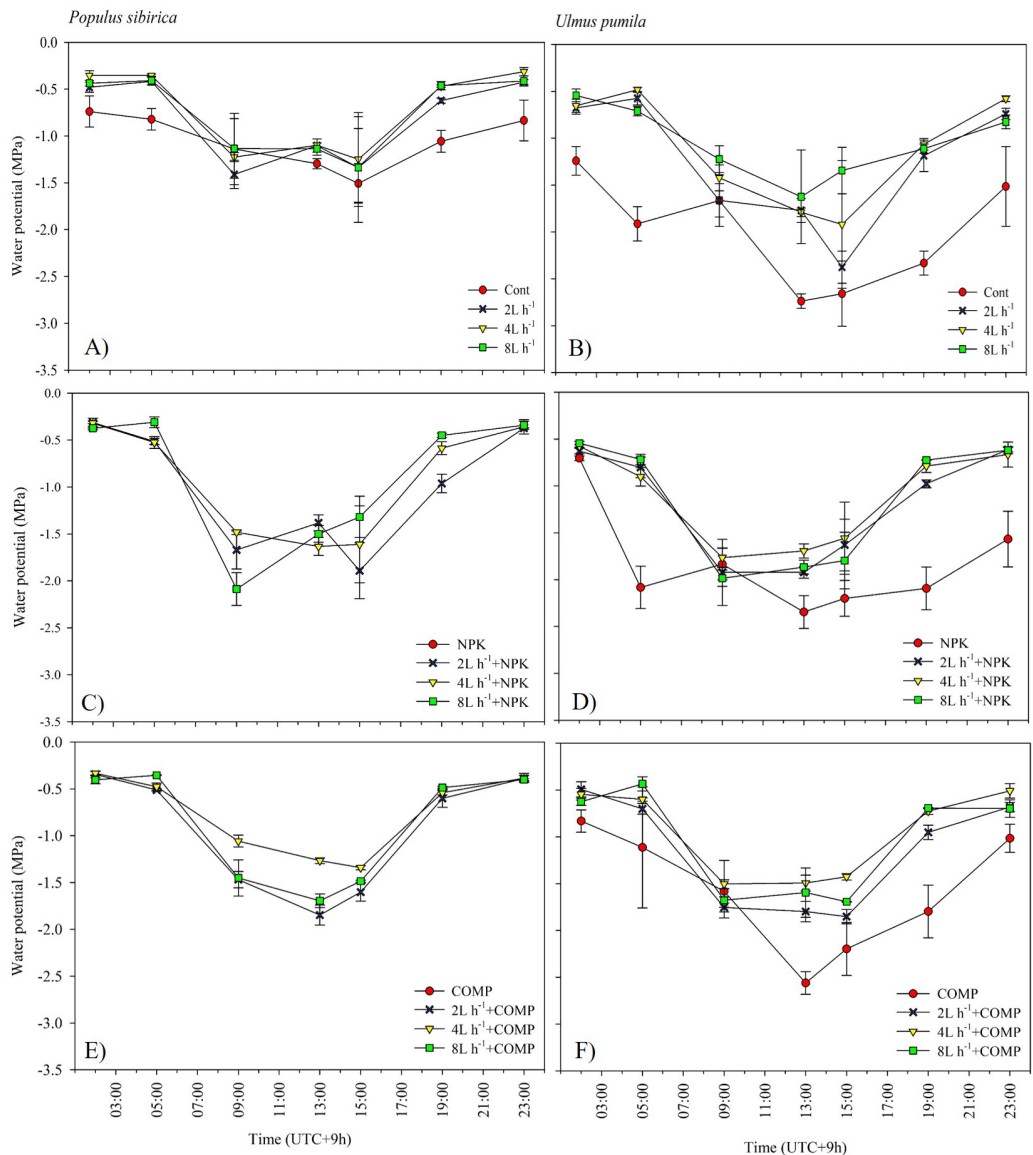

**Figure 7 Diurnal variation in leaf water potential (Ψ) of *Populus sibirica* (A) only water treatments, (C) Water+NPK, (E) Water+COMP, and Ulmus pumila (B) only water treatments, (D) Water+NPK, (F) Water+COMP across different treatments measured in July 2021. .** Vertical bars represent standard errors.

season. The findings indicate that the combination of water with COMP and/or NPK is necessary to support the growth of the seedlings, especially in the second year after the initial growth stage, in order to achieve their maximum growth potential. This combination of water and fertilizers likely stimulated the activity and movement of biological organisms present in the compost, leading to improved soil aeration, moisture retention, and nutrient availability. Specifically, the application of organic fertilizer enhanced the nitrogen content in the soil, a crucial element that influences the vegetative development of plants by

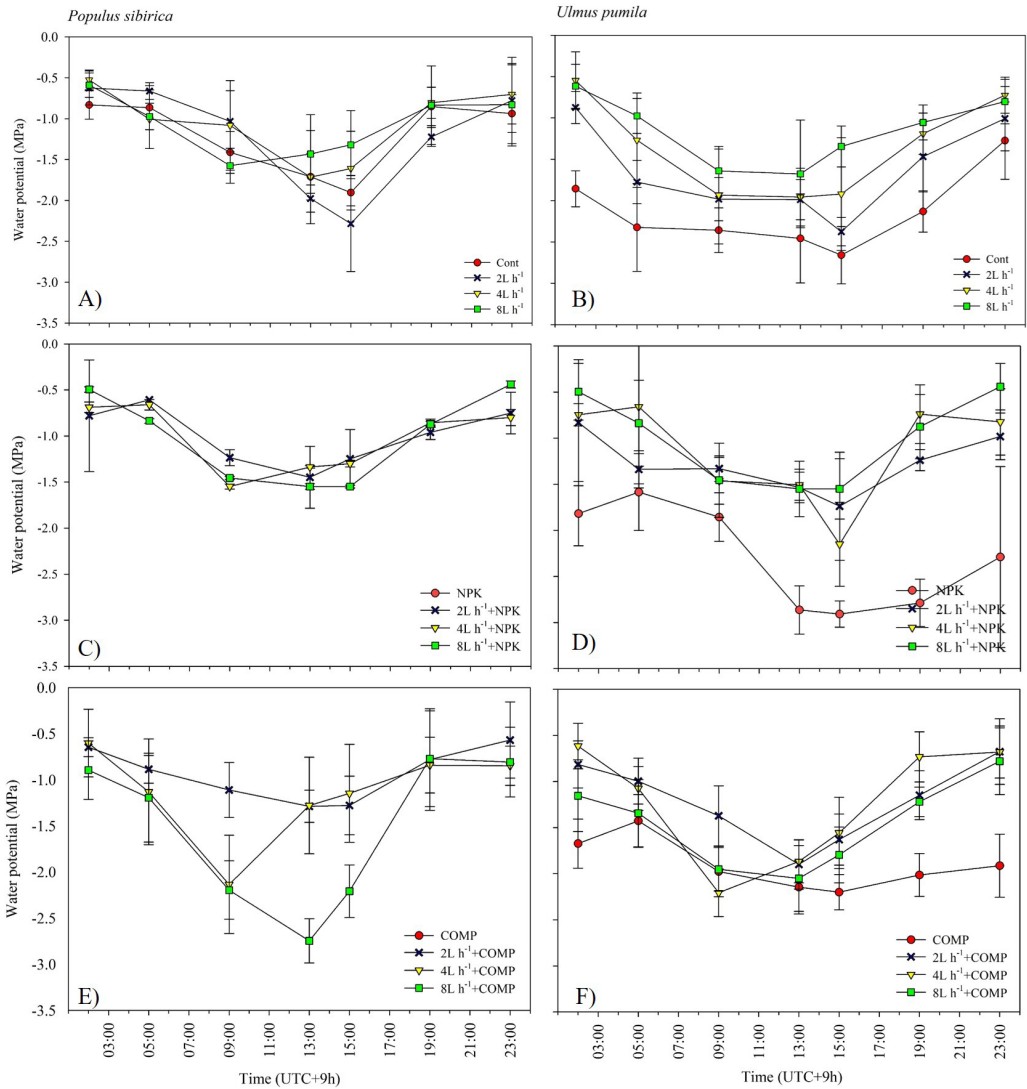

**Figure 8** **Diurnal variation in leaf water potential (Ψ) of *Populus sibirica* (A) only water treatments, (C) Water+NPK, (E) Water+COMP, and Ulmus pumila (B) only water treatments, (D) Water+NPK, (F) Water+COMP across different treatments measured in July 2022.** Vertical bars represent standard errors.

promoting increased leaf fresh weight, improved quality, and larger leaf area (*Mojeremane et al., 2015*). Furthermore, during the second growing season, the treatment of 2 L h⁻¹ + NPK resulted in a higher leaf area (LA) compared to other treatments in *P. sibirica*. This outcome can be attributed to the higher drought tolerance of *U. pumila*, where the addition of COMP did not provide significant benefits in terms of improving soil moisture and nutrient availability. Species with high invasiveness, like *U. pumila*, possess the ability to not only thrive in low-resource environments but also enhance macronutrient accumulation, soil properties, and soil microbial communities (*Souza-alonso & Novoa, 2014*; *Ye et al., 2019*). This explains the observed larger specific leaf area (SLA) of *U. pumila* in the control

Table 4 **Seasonal variation of the predawn ($\psi_p$) and midday ($\psi_m$) leaf water potentials in *Populus sibirica* and *Ulmus pumila* across different treatments measured in July 2021 and July 2022.** Different lowercase letters indicate significant differences among the treatments at $\alpha = 0.05$.

| Species | Treatments | $\psi_p$ | | $\psi_m$ | |
|---|---|---|---|---|---|
| | | July, 2021 | July, 2022 | July, 2021 | July, 2022 |
| *Populus sibirica* | CONT | $-0.82 \pm 0.11^c$ | $-0.86 \pm 0.56^{ab}$ | $-1.29 \pm 0.05^{abc}$ | $-1.71 \pm 0.53^{bc}$ |
| | 2 L h$^{-1}$ | $-0.41 \pm 0.08^{abc}$ | $-0.66 \pm 0.47^c$ | $-1.10 \pm 0.07^a$ | $-1.98 \pm 0.38^c$ |
| | 4 L h$^{-1}$ | $-0.35 \pm 0.01^{ab}$ | $-1.01 \pm 0.36^{abc}$ | $-1.09 \pm 0.02^a$ | $-1.72 \pm 0.19^{bc}$ |
| | 8 L h$^{-1}$ | $-0.40 \pm 0.04^{abc}$ | $-0.97 \pm 0.17^{abc}$ | $-1.53 \pm 0.09^{cd}$ | $-1.43 \pm 0.27^{ab}$ |
| | NPK | − | − | − | − |
| | 2 L h$^{-1}$+NPK | $-0.51 \pm 0.03^c$ | $-0.61 \pm 0.2^a$ | $-1.38 \pm 0.08^{bc}$ | $-1.45 \pm 0.35^{ab}$ |
| | 4 L h$^{-1}$+NPK | $-0.52 \pm 0.06^c$ | $-0.66 \pm 0.30^a$ | $-1.63 \pm 0.09^{de}$ | $-1.34 \pm 0.35^{ab}$ |
| | 8 L h$^{-1}$+NPK | $-0.31 \pm 0.05^a$ | $-0.83 \pm 0.46^{ab}$ | $-1.5 \pm 0.09^{bcd}$ | $-1.55 \pm 0.29^{abc}$ |
| | COMP | − | − | − | − |
| | 2 L h$^{-1}$+COMP | $-0.51 \pm 0.03^c$ | $-0.89 \pm 0.15^{bc}$ | $-1.85 \pm 0.10^e$ | $-1.28 \pm 0.17^{ab}$ |
| | 4 L h$^{-1}$+COMP | $-0.48 \pm 0.04^{bc}$ | $-1.13 \pm 0.57^{bc}$ | $-1.27 \pm 0.03^{ab}$ | $-1.28 \pm 0.52^{ab}$ |
| | 8 L h$^{-1}$+COMP | $-0.35 \pm 0.03^{ab}$ | $-1.19 \pm 0.48^{bc}$ | $-1.69 \pm 0.07^{de}$ | $-2.74 \pm 0.24^{bc}$ |
| *Ulmus pumila* | CONT | $-1.91 \pm 0.18^b$ | $-2.33 \pm 0.53^e$ | $-2.74 \pm 0.07^c$ | $-2.46 \pm 0.54^{bc}$ |
| | 2 L h$^{-1}$ | $-0.57 \pm 0.07^a$ | $-1.78 \pm 0.26^{de}$ | $-1.77 \pm 0.12^{ab}$ | $-1.99 \pm 0.24^{ab}$ |
| | 4 L h$^{-1}$ | $-0.48 \pm 0.02^a$ | $-1.27 \pm 0.57^{abcd}$ | $-1.79 \pm 0.04^{ab}$ | $-1.96 \pm 0.35^{ab}$ |
| | 8 L h$^{-1}$ | $-0.71 \pm 0.05^a$ | $-0.98 \pm 0.21^{abc}$ | $-1.62 \pm 0.49^a$ | $-1.68 \pm 0.65^a$ |
| | NPK | $-2.08 \pm 0.22^b$ | $-1.59 \pm 0.26^{dc}$ | $-2.35 \pm 0.17^{bc}$ | $-2.86 \pm 0.28^c$ |
| | 2 L h$^{-1}$+NPK | $-0.80 \pm 0.10^a$ | $-1.34 \pm 0.10^{bcd}$ | $-1.92 \pm 0.06^{ab}$ | $-1.54 \pm 0.16^a$ |
| | 4 L h$^{-1}$+NPK | $-0.90 \pm 0.02^a$ | $-0.66 \pm 0.35^a$ | $-1.79 \pm 0.04^a$ | $-1.51 \pm 0.56^a$ |
| | 8 L h$^{-1}$+NPK | $-0.71 \pm 0.05^a$ | $-0.83 \pm 0.16^{ab}$ | $-1.86 \pm 0.07^{ab}$ | $-1.55 \pm 0.48^a$ |
| | COMP | $-1.11 \pm 0.64^a$ | $-1.43 \pm 0.28^{bcd}$ | $-2.56 \pm 0.12^c$ | $-2.15 \pm 0.28^{ab}$ |
| | 2 L h$^{-1}$+COMP | $-0.70 \pm 0.05^a$ | $-0.99 \pm 0.25^{abc}$ | $-1.79 \pm 0.11^{ab}$ | $-1.89 \pm 0.26^{ab}$ |
| | 4 L h$^{-1}$+COMP | $-0.60 \pm 0.01^a$ | $-1.08 \pm 0.24^{abc}$ | $-1.49 \pm 0.08^a$ | $-1.87 \pm 0.23^{ab}$ |
| | 8 L h$^{-1}$+COMP | $-0.43 \pm 0.07^a$ | $-1.35 \pm 0.36^{bcd}$ | $-1.59 \pm 0.26^a$ | $-2.05 \pm 0.35^{ab}$ |

treatment and 4 L h$^{-1}$ without fertilizer during the first and second growing seasons, respectively.

## Effects of different fertilization and watering regimes on physiological traits of *Populus sibirica* and *Ulmus pumila*

In this study, the leaf physiological traits of both species were significantly influenced by different water and fertilizer treatments, which aligns with the findings of previous experiments (*Sumaira, Tao & Zhou, 2017*; *Byambadorj et al., 2021b*; *Byambadorj et al., 2021a*; *Guo et al., 2021*). Interestingly, higher chlorophyll concentration was observed in the control treatment, all treatments with water only, and the 2 L h$^{-1}$ + NPK treatment of *P. sibirica* seedlings compared to the other treatments, particularly during the first growing season. Similarly, the 2 L h$^{-1}$ treatment without any fertilizer resulted in a higher chlorophyll concentration compared to the other treatments in *U. pumila*. Despite the semi-arid conditions at the study site, our results indicate that both species are capable of synthesizing chlorophyll and maintaining high levels in their leaves even without the addition of fertilizer or irrigation. This is noteworthy considering the significantly

lower fluorescence parameters ($PI_{ABS}$ and $F_m$) observed in the control treatment. While many studies have reported a significant decline or alteration in chlorophyll content in plants subjected to long-term drought stress, most of these studies focused on herbaceous plants (*Li et al., 2006*; *Khayatnezhad & Gholamin, 2012*; *Chen et al., 2016*). Therefore, the observed stability of chlorophyll content in *P. sibirica* and *U. pumila* could serve as a potential indicator of their adaptation to drought conditions.

The current study also revealed different patterns in $g_s$ depending on the growing season and species, which can be attributed to the inherent adaptation of the species to the prevailing environmental conditions at the study site. In the case of *P. sibirica*, $g_s$ was significantly higher in the control treatment (CONT) during the early growth stage. This higher $g_s$ in CONT can be attributed to the smaller leaf area of seedlings under this treatment, leading to increased exposure of leaves to higher solar radiation and air temperature. Previous research has shown that despite a decrease in leaf water potential, stomatal conductance of *Populus* and *Pinus* species increases with rising temperatures (*Urban et al., 2017*). However, prolonged increases in $g_s$ can result in substantial water loss through transpiration. This explains the generally lower (more negative) predawn and midday leaf water potential observed in *P. sibirica* under CONT. It is likely that the species avoids excessive water loss by reducing leaf area during the early growth stage. Smaller leaves with fewer stomata can minimize water loss through transpiration (*Wang et al., 2019*; *Hernandez & Park, 2022*). This also explains the higher $g_s$ observed in the 2 L $h^{-1}$ + NPK treatment during the second growing season. As compared to the early growth stage in the first season, the demand for water and nutrients may have increased as the seedlings grew rapidly. Higher $g_s$ may have allowed growing leaves to maximize photosynthetic exchange and water conductance before senescence, which can be accelerated by abiotic stresses such as drought, nutrient limitation, extreme temperatures, and oxidative stress commonly experienced in dry land environments.

In the case of *U. pumila*, the treatment with NPK/COMP fertilizers without irrigation resulted in the highest $g_s$ among the treatments, despite lower $\psi_p$ and $\psi_m$. This finding supports the results of *Cavaleri & Sack (2010)*, who reported that invasive species exhibit higher stomatal conductance and sap flow rates compared to native species. The application of fertilizer may have played a role in regulating stomatal movement in *U. pumila*, leading to increased $g_s$ even in the dry environment of the study site. For instance, a study demonstrated that both mesophyll and stomatal conductance were reduced in seedlings with potassium deficiency (*Jin et al., 2011*). Potassium deficiency can cause dryness throughout the plant and hinder shoot development and water-use efficiency (*Arquero, Barranco & Benlloch, 2006*). Therefore, the observed high $g_s$ in *U. pumila* under the NPK/COMP treatment without irrigation may be attributed to a dehydration coping mechanism that helps maintain cell turgidity despite decreasing leaf water potential, thereby avoiding excessive water loss through transpiration.

### Implications for effective selection of species potentially suitable in dry lands

When considering the selection of tree species for establishing plantations in semi-arid and arid regions, where climate conditions are projected to become warmer and drier, it is generally preferable to choose species that can conserve water. In this study, both *P. sibirica* and *U. pumila* exhibited distinct responses to the irrigation and fertilization treatments, indicating their adaptability to thrive in arid regions with limited water availability. These species have demonstrated characteristics of water conservation, suggesting that they may also be well-suited to nutrient-poor soils and can tolerate low nutrient levels. Specifically, both species displayed the ability to efficiently allocate water and nutrient resources by adjusting leaf area growth, chlorophyll concentration, and stomatal conductance in response to different irrigation and fertilizer levels. This adaptive behavior allows them to optimize photosynthetic exchange and water conductance, even under conditions of limited water and nutrient availability.

The observed higher $PI_{ABS}$ and $F_m$ in all treated seedlings of both species, regardless of fertilizer type and amount of irrigation, can support such maximization of photosynthetic exchange and water conductance, although those with lower amount of irrigation are much preferable given the arid conditions in the area. Higher $PI_{ABS}$ suggests a stable, if not enhanced, overall photosynthetic performance, in treated seedlings of both species. Higher $PI_{ABS}$ can also be associated with the observed larger LA in treated seedlings (2 L $h^{-1}$ without fertilizer and 2 L $h^{-1}$ + NPK) because it also promotes development of photosynthetically active plant organs (*Lepeduš et al., 2011*). Monitoring $PI_{ABS}$ of *P. sibirica* and *U. pumila* throughout their growing period can provide valuable insights into a plant's response to drought stress and its adaptive mechanisms.

The response of *U. pumila* to the COMP fertilizer appeared to be more favorable compared to NPK, regardless of the presence or absence of irrigation. On the other hand, *P. sibirica* showed a higher degree of responsiveness to NPK when coupled with irrigation. Therefore, the suitability of these species in a specific semi-arid or arid environment may depend on the initial levels of soil water and nutrients. It is important to assess the initial conditions of soil moisture and nutrient content in the target planting area, or similar environments, prior to introducing seedlings of *P. sibirica* and *U. pumila*. This step is crucial to ensure long-term sustainability and achieve a high rate of plant survival.

## CONCLUSION

This study aimed to evaluate the impact of different watering and fertilizer treatments on the leaf morpho-physiological characteristics of both *P. sibirica* and *U. pumila* seedlings. The control seedlings, which received no irrigation or fertilizer, exhibited greater negative effects and less desirable traits compared to the treated seedlings. Despite the known adaptability of these species to various environmental conditions, our findings demonstrated that their leaf morpho-physiological traits were significantly influenced by the applied watering and fertilizer treatments. The extent of these effects varied depending on the growing season, irrigation amount, and fertilizer type. Generally, the application of 2–4 L $h^{-1}$ of water,

with or without fertilizer, was found to enhance the leaf morpho-physiological responses of both species, while adding 8 L h$^{-1}$ of water did not lead to improved performance.

## ACKNOWLEDGEMENTS

We thank the members of the Laboratory of Forest Genetics and Ecophysiology, National University of Mongolia for their help in field survey and plant material collections.

### Funding

The authors received no funding for this work.

### Competing Interests

The authors declare there are no competing interests.

### Author Contributions

- Ser-Oddamba Byambadorj conceived and designed the experiments, performed the experiments, analyzed the data, authored or reviewed drafts of the article, and approved the final draft.
- Jonathan Ogayon Hernandez analyzed the data, authored or reviewed drafts of the article, and approved the final draft.
- Sarangua Lkhagvasuren performed the experiments, prepared figures and/or tables, and approved the final draft.
- Ge Erma performed the experiments, prepared figures and/or tables, and approved the final draft.
- Khulan Sharavdorj performed the experiments, analyzed the data, prepared figures and/or tables, and approved the final draft.
- Byung Bae Park analyzed the data, authored or reviewed drafts of the article, and approved the final draft.
- Batkhuu Nyam-Osor conceived and designed the experiments, performed the experiments, authored or reviewed drafts of the article, and approved the final draft.

### Data Availability

The raw data are available in the Supplemental Files.

### Supplemental Information

Supplemental information for this article can be found online at http://dx.doi.org/10.7717/peerj.16107#supplemental-information.

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
