# Peer review of "Leaf morpho-physiological traits of Populus sibirica and Ulmus pumila in different irrigation regimes and fertilizer types"

_PeerJ, doi:10.7717/peerj.16107_

## Round 0.1 · original submission · Major Revisions

Thank you for submitting your work to PeerJ. I have received the reviews from three referees. All reviewers found the topic is important, but they also raised some major concerns regarding the experimental design, statistical method, data interpretation, and English writing. We hope these comments will improve the manuscript going forward.

Reviewer 1 ·

Basic reporting

This manuscript investigated the morpho-physiological responses of two fast-growing tree species, i.e., Populus sibirica and Ulmus pumila, to different irrigation regimes and fertilizer types to explore the optimal water and nutrient requirements of these two species. This is an important topic in tree physiology and can shed light on afforestation in drylands. Although a lot of data were present here, I have some major concerns regarding the experimental design, statistical method and data interpretation. Below are some general and specific comments for the improvement of the manuscript.


General comments
1. While a 2.5 m distance was retained between experimentally treated plots, I wonder whether the exchange of soil moisture between plots with contrasting water treatments was completely eliminated. This can be easily achieved by vertically installing some water-proofing plates/films between blocks. Besides, were soil water content and nutrients measured synchronously during the two growing seasons when physio-morpho traits were tested?

2. Given that three factors influencing plant performance were involved in this study, I believe that using one-way ANOVA as the main statistical analysis technique is not appropriate. I suggest using linear mixed effect models to test the effects of predictive factors on responsible variables by assigning water availability, fertilizer type and growing season as fixed main effects and individual plants as the random effect. At least, multiple-way ANOVA should be performed to test the effects of water availability, fertilizer type, growing season and their interactions on plant performance.

3. The reasons why the physio-morpho measurements were done in two consecutive growing seasons need to be better clarified. Were there significant differences in meteorological conditions between these two years, or were you just treating the two-year measurements as two duplicates? Also, the finding of significant differences between growing seasons needs to be better explained, if the meteorology was comparable, why trees showed inter-annual variations in physio-morpho characteristics?

4. The inter-specific difference in responding to contrasting water and nutrient regimes needs to be better explored and highlighted. Which one is more recommended for establishing tree plantations in this environment where the warming and drying climate is increasingly anticipated?

Line-specific comments:
line 37: CONT
lines 173, 180 and 191: Please double-check the sampling size (n) for each physiological trait measured.
line 232: Table 3?
lines 297-299: This may be not true, please tone down this argument.
lines 304-305: Relevant references supporting this point are needed to be added here.
line 327: This species-specific pattern could be attributed to...
line 337: supports
Figures 1 and 2 were not cited in the main text.

Experimental design

no comment here, please refers to the comments above.

Validity of the findings

no comment here, please refers to the comments above.

Additional comments

no comment here, please refers to the comments above.

Reviewer 2 ·

Basic reporting

The study investigated the morpho-physiological responses of Populus sibirica and Ulmus pumila fast-growing tree species to different watering regimes and fertilizer types to elucidate their water and nutrient requirements. The study found that the two species responded differently to the different watering and fertilizer treatments.

The experiment was well designed and abundant data was collected. However, the authors did not clearly describe how the experiments were conducted. Additionally, the authors presented the same data both in tables and figures, which is redundant. The English language in the study could also be improved, especially in long sentences. The authors should proofread the study carefully to ensure that they clearly express what they want to say.

Experimental design

1. Lines 43-45: This sentence needs to be rewritten. As well as lines 45-47.
2. Line 142: What do authors mean by ‘acclimatization period’?
3. Lines 137-138: According to the materials and methods, trees were transplanted 10 years ago. Then when were the nutrient and irrigation treatments applied? Authors mentioned fertilizers were filled in the transplanting holes, so was the fertilization treatment applied 10 years ago? It doesn’t make sense.
4. The timeline of the experiments in this study is confusing, can authors please make it clear? Authors should mention the experiment started from which month of which year; data was collected on which day.
5. Line 170: Please explain LDM.

Validity of the findings

6. Table 3: Authors may only present Fv/Fm and PIabs since the title of table 3 only included these 2 parameters. It seems authors didn’t discuss the chlorophyll fluorescence parameters results. However, chlorophyll fluorescence is sensitive to drought stress, they should be well discussed. And authors should introduce these parameters, explain what meanings they can tell instead of just presenting these data and comparing which is bigger or smaller.
7. For many parameters, if authors decide to present them in tables, it’s not necessary to present them again in figures. Authors have done enough work to conduct the experiments, collect data, and make good tables/figures. You don’t have to present results repeatedly.

·

Basic reporting

The manuscript “Leaf morpho-physiological traits of Populus sibirica and Ulmus pumila in different irrigation regimes and fertilizer types” involves the impacts of irrigation and fertilizer management on leaf morpho-physiological trait in two kinds of trees. The results seem to be confidential and valuable. However, there are many redundant sentences and professional terms and complex sentences in the text, which needs to be revised. Authors tend to use conjunctions of causation, but a lot of places actually are not required. So the manuscript needs careful editing by someone with expertise in writing and reference format. Some major concerns should be solved before publication, as elaborated below.
Abstract
L36 what does COMP mean? should be explained clearly.
L36-48 These reuslts are boring when reading, and the authors should provide the signicant results for elucidating your hypothesis rather than listing your findings.
Introduction
L114-117 The hypothesis should be listed here?

Experimental design

Methods
Two different species were involved in this study, what's the differnce between them and why not one or three?
These amounts seems to be overuesed, why did you design these rate? And the authors did not consider 0 g/tree? This means the authors design a two-factor experiment that include the interacted tratement,
The authors did not consider none fertilzer treatment, why did you say here?

Validity of the findings

Results
L229-290 The results seems to be vert sweeping, and better reorganize paras structure and split three and more parts to list your signicant findings.
Discussion
“Effects of different fertilization and watering regimes on the leaf area and specific leaf area” – This parts about the leaf area and specific leaf area, and the authors should focus on important results and differences in morphological traits between this study and previous research.
please insert an additional subsection with an explicit subtitle discussing the main implications of your findings for assessing agronomic practices beyond your specific case study. This out-scaling is important for maximising appeal to our global readership.

Additional comments

References
Here, the authors should double-check the formats with the criteria of PeerJ. Many errors in the present version.

---

## Round 0.2 · Minor Revisions

Most of the concerns raised by reviewers have been addressed. The manuscript has been improved a lot. However, there are still some issues needed to be addressed. Please see the reviewers' comments.

Reviewer 1 ·

Basic reporting

This is an improved version of a manuscript that I recently reviewed. The authors have carefully responded to each of the raised comments. The speculative tone of some statements has been moderated. Methodological and statistical doubts have been clarified. I am happy to see that the authors now well discuss the implication of studying tree stress physiology for afforestation in drylands. Now, the manuscript is ready to be published in PeeeJ.

Experimental design

no comment

Validity of the findings

no comment

Additional comments

no comment

Reviewer 2 ·

Basic reporting

I appreciate the extensive revisions made by the authors in response to my comments. They have addressed most of my concerns. However, there are a few answers to my questions that were only included in the rebuttal letter, but not in the manuscript itself. I recommend that the authors add these explanations to the manuscript as well, as other readers may have similar questions.

'3. Lines 137-138: According to the materials and methods, trees were transplanted 10 years ago. Then when were the nutrient and irrigation treatments applied? Authors mentioned fertilizers were filled in the transplanting holes, so was the fertilization treatment applied 10 years ago? It doesn’t make sense.

Response: We are sorry for the confusion. Irrigation was applied every 4 days and fertilizer was applied once in every two years.
4. The timeline of the experiments in this study is confusing, can authors please make it clear? Authors should mention the experiment started from which month of which year; data was collected on which day.

Response: The timeline of the experiments was specified in the materials and methods of the manuscript. We started the experiment in mid-May 2011 and we started to measure leaf morpho-physiological traits for this study from June to September of 2021 and 2022. The present study is one of the components of a big project in Mongolia, so the experiments were already 12 years old. '
I am pleased to inform you that after the last round of revision, the manuscript has been improved a lot, and it can be accepted for publication.

Experimental design

no comment

Validity of the findings

no comment

·

Basic reporting

The current revision has been revised carefully, and the issues has been addressed. However, I have minor comments on the revision before publication as following:
1) Format errors with nonitalic in Latin name of species: L809, L836, L849, L855, L868
2) Page or DOI error: L790

Experimental design

no

Validity of the findings

no

Additional comments

no

---

## Round 0.3 · accepted · Accept

The current manuscript has been revised carefully, and is ready for publication.